# Variability in ‘Capri’ Everbearing Strawberry Quality during a Harvest Season

**DOI:** 10.3390/foods12061349

**Published:** 2023-03-22

**Authors:** Kristyna Simkova, Robert Veberic, Metka Hudina, Mariana Cecilia Grohar, Tea Ivancic, Tina Smrke, Massimiliano Pelacci, Jerneja Jakopic

**Affiliations:** Department of Agronomy, Biotechnical Faculty, University of Ljubljana, Jamnikarjeva 101, SI-1000 Ljubljana, Slovenia

**Keywords:** strawberry, fruit quality, harvest time, temperature, sunshine duration

## Abstract

Strawberries are appreciated by consumers for their characteristic taste and health benefits, which enhance their demand throughout the year. Everbearing strawberries can produce fruits for a longer period and could thus meet this demand, but the fruit quality depends on environmental factors and the cultivar. This study focused on the effect of environmental conditions on the physical attributes and the composition of everbearing Capri cultivar fruit harvested from the end of June to the end of October. A positive correlation was observed between temperature and organic acid content (r = 0.87), and a positive correlation was observed between sunshine duration, anthocyanin (r = 0.87) and phenolic compound contents (r = 0.89). Additionally, the composition of sugars was affected by the environmental conditions. While strawberries harvested towards the end of October, when lower temperatures predominated, were larger in size and had a higher sugar/acid ratio, fruit harvested in the middle of August, when there were longer periods of sunshine, had higher anthocyanin and phenolic compound contents. In conclusion, strawberries with higher sugar/acid ratios are obtained when temperatures are lower, while strawberries exposed to longer periods of sunshine are richer in health-promoting compounds.

## 1. Introduction

Strawberry is one of the most popular fruits, with its worldwide production having reached approximately 12.2 million tons in 2020 [1]. Strawberries are appreciated by consumers for their characteristic flavour, which is attributed to the ratio of sugar to acid content combined with the volatile compound profile [2]. Strawberries are also valued due to an extremely high amount of vitamin C, which increases their nutritional importance [3,4]. Together with vitamin C, strawberries are also a source of other important micronutrients, such as folate, and, in lesser amounts, thiamin, riboflavin, niacin, and vitamins B6, K, A and E [4]. Apart from vitamins, strawberries contain minerals such as manganese, potassium, magnesium, copper, iron, and phosphorus [4]. Moreover, strawberries accumulate a variety of phenolic compounds, such as flavonoids (mainly anthocyanins) and phenolic acids (hydroxybenzoic and hydroxycinnamic acids) [5]. Additionally, according to epidemiological studies, phenolic compounds are responsible for lowering the risk of chronic diseases such as cancer and cardiovascular diseases [6,7,8]. These health benefits make strawberries an important source of nutrients and can further enhance the consumer demand. The consumer demand in Europe has increased all year round, and there is a growing interest in everbearing cultivars, since they can produce fruits for a longer period, from late spring to autumn [9,10]. While June-bearing strawberry genotypes have been extensively studied, environmental regulations and the impact of environmental conditions on fruit quality and composition have been less studied in everbearing genotypes. Previous studies on everbearing strawberries [10,11] have mainly focused on flowering and yield performance under different environmental conditions, but there are few studies focusing on the changes in the fruit composition of everbearing strawberries [12,13].

The quality of strawberry fruit depends on many factors, including the cultivar, growing location, climatic conditions during growth and harvest, time and method of harvest, ripening stage and postharvest handling [14]. According to previous works [15,16,17], the genotype largely determines the organoleptic and functional fruit quality parameters (acidity, sugar content and phenolic content), but within the same cultivar, the contents vary at different harvest timepoints. Among the environmental factors that influence strawberry plant performance are temperature and sunshine duration [18,19].

Based on previous studies, strawberry total sugar content increases [20], but sugar content tends to decrease, towards the end of the season [13], since the photosynthetic capacity of these plants is lower due to the decrease in sunshine [13]. These changes depend on the photosynthetic rate, which, in general, increases as the temperature increases, but once it exceeds the optimal value for plant growth, photosynthesis is inhibited, and photorespiration may occur [21]. Higher temperatures could result in increased respiration but can also cause lower contents of organic acids [13]. However, the opposite trend was also previously reported [22,23].

Additionally, the synthesis of phenolic compounds is affected by different environmental factors [24]. Both light and temperature are important factors affecting the flavonoid pathways in fruits [25,26], which are responsible for the biosynthesis of the main pigments of strawberries, i.e., anthocyanins and other flavonoids. It was previously reported that anthocyanin accumulation is higher in strawberries grown under higher temperatures and with a longer sunshine duration [12,27,28]. However, some anthocyanins can be less affected by environmental conditions [27].

Overall, the combined effect of genotype and environmental interactions determines fruit quality and, consequently, consumer acceptance. Therefore, understanding these interactions is necessary to find the optimal conditions for each selected cultivar in order to obtain fruit of good organoleptic and nutritional quality.

The objective of this study was to determine the optimal harvest conditions for the common everbearing Capri strawberry cultivar, commercially grown in Slovenia. Our aim was to study the impact of environmental conditions during the harvest season on the physical and chemical attributes of strawberry fruit, with a focus on sugars, organic acids and phenolic compounds. Additionally, ascorbic acid was analysed as the major micronutrient in strawberries. The environmental conditions (temperature, relative humidity and sunshine duration) were closely monitored throughout the season in the field so that their effect on fruit quality could be evaluated.

## 2. Materials and Methods

### 2.1. Plant Material

The experiment was carried out in Pesje, southeast Slovenia (latitude 45°56′26″ N, longitude 15°33′11″ E), in an open field with an integrated production system equipped with a drip irrigation system. The experimental design had five blocks with sixty plants per block. The cultivar selected for this study was the everbearing Capri cultivar, planted in autumn 2020. This strawberry cultivar was chosen because it is commonly grown in Slovenia, provides a high yield and has strong disease resistance [29]. Strawberry fruit was collected bi-weekly during the period between the end of June and the end of October 2021. A total of 10 harvests were evaluated.

With each sampling, commercial strawberries were randomly picked from the whole plot, and only fruit that appeared technologically ripe was selected for this experiment. Technologically ripe fruit was fully red, including the area around the calyx, and was fit to be sold for fresh consumption based on the producer’s experience. With each harvest, 1 kg of strawberries was obtained for analysis. Additionally, fruit was sorted based on colour to eliminate fruits that may have been unripe or overripe (Appendix A). For the measurement of physical parameters, 15 representative fruits were chosen and individually measured. For chemical analysis, 5 biological repetitions were prepared by pooling at least 15 strawberry fruits from the whole plot.

Temperature and relative humidity (RH) data were collected hourly with sensors (Voltcraft DL-121TH; Hirschau, Germany) in the fields. Data on sunshine duration were obtained from Slovenian Environment Agency (ARSO) at the meteorological station in Novo Mesto.

### 2.2. Physical Measurements

Strawberries were sorted by colour and freedom from defects. Upon each collection, fifteen fruits of the same colour and size were randomly picked for the measurement of colour, size, weight, total soluble solids (TSSs), firmness and index of ripeness.

The weight of each fruit was noted. For size determination, the length and diameter of each fruit were measured with a digital ruler. The length was measured from the calyx to the apex of the fruit, and the diameter was measured at the widest part of the fruit. The colour of the fruits was measured with a colourimeter (CR-10 Chroma; Minolta, Osaka, Japan). The colour parameters were measured in the CIELAB colour space, where the L* value corresponds to lightness (0 is black, and 100 is white); the h° value corresponds to colour expressed in degrees (0° is red; 90° is yellow; 180° is green; and 270° is blue); and the C* value corresponds to chroma (a higher value means more intense colour). TSSs were measured with a refractometer (MA885 Wine Refractometer; Milwaukee, WI, USA). Firmness was measured with a digital penetrometer (TR Turoni, Turin, Italy) with a 3 mm plunger. The index of ripeness of fruit was measured with a DA meter FRM01F (Sintéleia, Bologna, Italy), which measures the amount of chlorophyll inside the fruit using its absorbance.

### 2.3. Dry Matter

For each harvest, strawberry dry matter was determined in an amount of approximately 10 g, pooled from at least 15 fruits, which was placed in paper bags in 5 repetitions and dried in an oven at 110 °C for 3 days. Subsequently, the percentage of dry weight was calculated, and these results were used for the calculation of chemical composition per dry weight.

### 2.4. Sample Preparation

Fruits were cut in quarters, and the pieces were randomly separated. Five replicates (approx. 10 g each) were prepared for each extraction procedure.

### 2.5. Ascorbic Acid Extraction and Determination

Extraction was performed using fresh samples in five repetitions following the procedure described by Mikulic-Petkovsek et al. [30]. The sample (2.5 g) was extracted with 5 mL of 3% metaphosphoric acid and shaken for 30 min at room temperature. After extraction, the samples were centrifuged for 10 min at 7000× *g* at 4 °C (Eppendorf Centrifuge 5810 R; Hamburg, Germany). The supernatant was then filtered using 0.20 µm cellulose filters (Macherey-Nagel, Düren, Germany). Samples were stored at −20 °C until analysis.

Samples were analysed using Vanquish HPLC (ThermoScientific, Waltham, MA, USA). The conditions of analysis were as follows: column (Rezex ROA-Organic acid H+ 8% (150 mm × 7.8 mm); Phenomenex, Torans, CA, USA) at a temperature of 20 °C and a flow rate of 0.6 mL min^−1^, with 4 mM sulphuric acid in bi-distilled water as mobile phase. The injection volume was 20 µL. The response of samples was measured with a UV detector at 245 nm. Ascorbic acid was identified using an external standard from Sigma-Aldrich (Steinheim, Germany).

### 2.6. Sugar and Organic Acid Extraction and Determination

The extraction and identification of sugars and organic acids followed the method previously described by Mikulic-Petkovsek et al. [31]. The samples were chopped, and 1 g was extracted with 1 mL of bi-distilled water and shaken for 30 min. After extraction, the samples were centrifuged for 10 min at 10,000× *g* at 4 °C (Eppendorf Centrifuge 5810 R; Hamburg, Germany). The supernatant was then filtered using 0.20 µm cellulose filters (Macherey-Nagel, Düren, Germany). Samples were stored at −20 °C until HPLC analysis.

Organic acids were analysed using Vanquish HPLC (ThermoScientific, Waltham, MA, USA). The conditions of analysis were as follows: column (Rezex ROA-Organic acid H+ 8% (150 mm × 7.8 mm); Phenomenex, Torans, CA, USA) at 65 °C and a flow rate of 0.6 mL min^−1^, with 4 mM sulfuric acid in bi-distilled water as mobile phase. The injection volume was 20 µL. The response of samples was detected with a UV detector at 210 nm. Organic acids were identified using external standards for citric, malic and fumaric acids from Fluka Chemie (Buchs, Switzerland) and shikimic acid from Sigma-Aldrich (Steinheim, Germany). The results were expressed as mg g^−1^ dry weight (DW).

Individual sugars were analysed using Vanquish HPLC (ThermoScientific, Waltham, MA, USA). The conditions of analysis were as follows: column (Rezex RCM-monosaccharide Ca+ 2% (300 mm × 7.8 mm); Phenomenex, Torans, CA, USA) at 65 °C and a flow rate of 0.6 mL min^−1^, with bi-distilled water as mobile phase. The injection volume was 20 µL. Individual sugars were identified using external standards for fructose, glucose and sucrose (Fluka Chemie GmBH, Buchs, Switzerland). The results were expressed as mg g^−1^ dry weight (DW).

### 2.7. Phenolic Extraction and Determination

Phenolic extraction was performed according to Mikulic-Petkovsek et al. [32] with some minor modifications. The sample (3 g) was extracted with 6 mL of 80% methanol acidified with formic acid (3%), put in a cooled ultrasonic bath (0 °C) for 1 h and then centrifuged for 10 min at 10,000× *g* at 4 °C (Eppendorf Centrifuge 5810 R; Hamburg, Germany). The supernatant was then filtered using 0.20 µm polyamide filters (Macherey-Nagel, Düren, Germany). Samples were stored at −20 °C until HPLC analysis.

Phenolic compound composition was analysed using a Dionex UltiMate 3000 HPLC (Thermo Scientific, Waltham, MA, USA) system. Spectra were measured at 280, 350 and 530 nm. The flow rate of the system was 0.6 mL min^−1^, and the injection volume of the samples was 20 μL. The mobile phases used were 3% acetonitrile and 0.1% formic acid in bi-distilled water (*v*/*v*/*v*), as mobile phase A, and 3% bi-distilled water and 0.1% formic acid in acetonitrile (*v*/*v*/*v*), as mobile phase B. The linear gradient used was 5% of solvent B from 0 to 15 min, 5–20% of solvent B from 15 to 20 min, 20–30% of solvent B from 20 to 30 min, 30–90% of solvent B from 30 to 35 min, 90–100% of solvent B from 35 to 45 min and then 100–5% of solvent B from 45 to 50 min.

Phenolic compounds were identified using the comparison with the standard retention times and using an LTQ XL mass spectrometer (Thermo Scientific, Waltham, MA, USA) based on their fragmentation pattern. The sample injection volume was 10 μL, and other chromatographic conditions were the same as those described for HPLC analyses. The mass spectrometer was operated in both negative and positive ion modes with electrospray ionisation (ESI). The capillary temperature was 250 °C, with sheath gas at 20 units and auxiliary gas at 8 units. The source voltage used was 4 kV, with *m/z* scanning from 115 to 1600. The quantification of phenolic compounds was performed according to a corresponding external standard or chemically similar compounds and expressed as mg 100 g^−1^ DW.

### 2.8. Enzyme Activity Measurements

Enzyme extraction and assays followed the procedure by Cebulj et al. [33] with minor modifications, as specified below.

#### 2.8.1. Extraction of Enzymes

For this extraction procedure, fruits were cut in quarters, shock-frozen with liquid nitrogen and stored at −80 °C. The fruits were ground with an IKA A11 basic grinder (IKA-Werke, Staufen, Germany) at a low temperature using liquid nitrogen. The sample (1 g) was mixed with 0.5 g of Polyclar and 4 mL of extraction buffer (0.01 M TRIS, 0.007 M EDTA and 0.01 M Borax). The samples were vortexed for 30 s and then centrifuged for 10 min at 10,000 rpm at 4 °C (Eppendorf Centrifuge 5810 R, Hamburg, Germany). The supernatant (400 µL) was then passed through a Sephadex G-25 gel column to remove low-molecular-weight compounds before measurement.

#### 2.8.2. POD and PPO Assays

For polyphenol oxidase (PPO) activity, the sample (130 μL) was mixed with 300 µL of McIlvaine buffer (0.1 M Na_2_HPO_4_) and 170 μL of 0.2 M pyrocatechol solution; then, absorbance was measured for 20 min at 410 nm.

For peroxidase (POD) activity, the sample (100 μL) was mixed with 1000 µL of H_2_O_2_ -KPi buffer and 10 μL of 0.04 M *o*-dianisidine solution in methanol; then, absorbance was measured for 20 min at 460 nm.

The measurements were performed with Genesys 10S UV-Vis Spectrometer (Thermo-Scientific, Waltham, MA, USA), and data were collected with VISIONlite software. Enzyme activity was expressed as U (units) per mg protein. One unit (U) is defined as the change in absorbance in one minute. The protein content was determined with the Bradford method with minor modifications in accordance with Kruger [34].

### 2.9. Statistical Analysis

The data were statistically analysed in R, version x64 4.1.2, using the Rcmdr graphical interface package, version 2.8-0. The data were expressed as means ± standard error. In order to determine significant differences among the data, one-way analysis of variance (ANOVA) was used with Tukey’s tests. Significant differences were considered at *p* < 0.05. To determine the correlations of physical and chemical parameters with environmental conditions (temperature and relative humidity), the Pearson correlation test was applied. In correlation tests, the average daily values of temperature and relative humidity, as well as sunshine duration, of the 7 days before the collection of fruit were considered.

## 3. Results and Discussion

### 3.1. Environmental Conditions

The daily average temperature and RH values are shown in Figure 1. During the experiment, the maximum average daily temperature was 27.9 °C (8 July). At this time, the lowest average RH was also reached, which was 59.8% (9 July). On the other hand, the highest RH was 99.5% (10 October), and the minimum average daily temperature was 6.1 °C (25 October).

Figure 2 shows the daily and weekly average sunshine duration from the middle of June to the end of October. The highest weekly average sunshine duration was 12.6 h, during the week from 21 June to 27 June, and the lowest weekly average was 3 h, during the week from 4 October to 10 October.

### 3.2. Physical Parameters

All parameters measured after each harvest are presented in Table 1. TSSs (total soluble solids) and the ripening index were constant during the harvest season, apart from a few exceptions. The TSS values were not lower than 7 °Bx, which is the recommended limit for acceptable flavour [35]. Small differences in the ripening index confirmed that all the fruits picked for measurement were in the same ripening stage.

Physical characteristics such as size and firmness are also important quality attributes for consumers and could affect the marketability of fruit. The size of fruits can be affected by the temperature during the growth period [36]. For the Capri cultivar, the collected fruits were smaller in size and weight when the temperature was higher. However, the average diameter did not decrease under 18 mm, which is considered the minimum, as defined by the UNECE standard [37]. The length of fruit showed a positive correlation with RH and a negative correlation with temperature. The weight of the strawberries significantly changed during the collection season and showed correlations similar to those relative to the length, reaching the highest weight at the beginning and the end of the season, i.e., in June and October, respectively. This is in agreement with previous studies where fruit weight was lower at higher temperatures when grown in a controlled environment in greenhouses [20,38]. Water stress was excluded as a factor from our study, since there was a stable irrigation system. On the other hand, there seemed to be no correlations between these parameters and sunshine duration.

The firmness of the fruit changed during the harvest season, with the firmest fruit having been picked at the end of the season (28 October) and the least firm fruit having been picked at the beginning of the season (23 June and 7 July). However, we did not find any correlation between firmness and environmental factors, which contradicts previous findings [12,16,39] that indicated that high temperatures had a negative impact on fruit firmness. However, firmness can also be influenced by other factors, such as the activity of degrading enzymes and cell wall composition [23,39].

Regarding colour, all the parameters showed a negative correlation with temperature. Additionally, C* (chroma) showed a positive correlation with RH and a negative correlation with sunshine duration, meaning that the colour intensity was higher at higher RHs and lower temperatures, and with shorter sunshine durations. A similar effect was also observed for the L* (lightness) parameter, meaning that strawberries were darker at lower temperatures and with shorter sunshine durations, but the differences were very small between the lowest (28.0) and the highest obtained values (32.4).

### 3.3. Ascorbic Acid Content

In strawberries, ascorbic acid is recognised as an essential hydrophilic micronutrient; therefore, its content is an important quality parameter. As seen in Figure 3, ascorbic acid content reached the maximum at the beginning of the season (23 June) and reached the minimum at the end of the season (27 October). Ascorbic acid content was positively correlated with both the average temperature and sunshine duration (84.8% and 83.2%, respectively; Table 2). Additionally, ascorbic acid content was negatively correlated with RH (−89.6%).

Although light is not necessary for the synthesis of ascorbic acid in plants, its amount and intensity during fruit development can influence the amount of ascorbic acid formed [40]. In our study, the strawberries harvested after days with longer sunshine durations showed higher levels of ascorbic acid. Significant differences in ascorbic acid content among different harvests were previously reported in different cultivars [12] and were attributed to the difference in light intensity. Moreover, it was reported that ascorbic acid content is enhanced by light intensity in tomatoes [41]. Its increased accumulation under high light intensity could reflect its use in H_2_O_2_ detoxification [42]. In our study, we observed a higher accumulation of ascorbic acid at higher temperatures, even though it was reported that high temperatures during the day (30–35 °C) have a negative effect on ascorbic acid content [28,40]. In our case, the highest average temperature did not exceed 28 °C, so it is suggested that temperature can have a positive effect on ascorbic acid content within a specific range. The effect of environmental factors on ascorbic acid synthesis is also dependent on the cultivar according to a previous study involving short-day strawberry cultivars that reported that ascorbic acid content did not show any correlation with the environmental conditions in most of the studied cultivars [16]. 

### 3.4. Organic Acid Contents

Total organic acid content is represented in Table 3. Total organic acid content reached the maximum in August and the minimum at the end of October. Moreover, a similar variation in content was observed in individual organic acids. Based on previous studies [13], higher temperatures could result in increased respiration and, consequently, in lower contents of organic acids. However, in our study, the results show that the environmental factors had the opposite effect on organic acid contents. Total organic acid content, as well as the individual contents of all detected organic acids, were correlated with the average temperature and RH (Table 2). On the other hand, no effects of sunshine duration were observed.

A positive correlation between titratable acidity and air temperature was previously reported by Agüero et al. [22] for strawberries grown under subtropical conditions, and a similar effect of temperature on acidity was also reported by Kannaujia and Asrey [23], whose study showed that the titratable acidity decreased as the temperature decreased, which was attributed to the lower accumulation and slower depletion of organic acids with respiration. In contrast, other studies indicated that organic acid content either was stable throughout the harvest season [43] or decreased at higher temperatures [20], which suggests that the effect of temperature on the contents of organic acids is also probably dependent on the cultivar.

### 3.5. Sugar Contents

Together with organic acids, sugar content is one of the most important quality attributes of strawberries, as they both greatly contribute to fruit flavour. There were no significant changes in total sugar content during the season (Table 4), a finding which is consistent with the values of total soluble sugars (Table 1). It was suggested that TSSs and total sugar content decrease towards the end of the season, since the photosynthetic capacity of plants is lower due to the decrease in sunshine [13]. In our study, this effect was not observed, as no correlations with the environmental conditions were detected, which was also previously reported for other cultivars [16,22].

Moreover, it is also important to look at the individual contents of different sugars, since the perception of sweetness of individual sugars can be different [44]. There were significant changes in individual sugar contents in our study (Table 4). The contents of glucose and fructose reached their maximum at the beginning of the season in June and their minimum at the end of October, whereas sucrose content reached the maximum value in October and the minimum value at the beginning of the harvest season, between the end of June and the beginning of August. Regarding the influence of environmental factors, a lower accumulation of sucrose was observed at higher temperatures and under a longer sunshine duration (Table 2). Additionally, the content of sucrose was positively correlated with RH. The synthesis of glucose and fructose contents increased under higher average temperatures and longer sunshine durations. Significant changes in the individual contents of different sugars were also observed by Ruan et al. [13] for different day-neutral and everbearing cultivars. These changes could be explained by the effect of environmental factors on the activity of enzymes involved in the sucrose/hexose interchange, such as sucrose-phosphate synthase, invertase and sucrose synthase [21].

### 3.6. Sugar/Acid Ratio

Among consumers, a high sugar/acid ratio is desired, which results from high sugar content or low acid content. As seen in Table 4, the sugar/acid ratio was the highest at the end of the harvest season (end of October) and the lowest in August. As mentioned above, total sugar content was stable during the season, so the ratio values were affected by the changes in organic acid content. The sugar/acid ratio was negatively correlated with the average temperature and was positively correlated with RH (Table 2), which corresponds with previous findings [43]. In the mentioned study, the changes were mainly caused by the changes in total sugar content, as sugar content decreased due to a lower photosynthetic rate. In our study, the changes were mainly caused by the changes in total organic acid content, not by those in sugar content.

### 3.7. Phenolic Contents

Strawberries contain various phenolic compounds, and their synthesis is affected by different environmental factors [24]. Phenolic compound content enhances the nutritional quality of fruit, but it also affects the colour of fruit, and it is an important quality parameter for juice processing [4,45,46,47]. The identified phenolic compounds are listed in Appendix A. Total phenolic content (including anthocyanin content) was positively correlated with RH (−77.2%) and sunshine duration (89.2%), but no correlations with temperature were observed (Table 5), which is in contrast to previous studies [12,28] where a higher accumulation of phenolic compounds was observed in strawberries grown at higher temperatures in greenhouses. However, the highest non-anthocyanin phenolic content values were reached in August and at the end of September, and the total content of non-anthocyanin phenolic compounds showed a positive correlation with temperature (69.4%). This suggests that it is the synthesis of non-anthocyanin phenolic compounds that is affected by temperature, rather than the content of anthocyanins. Most of the separate groups of phenolic compounds, except for flavonols, also showed increased synthesis at higher temperatures. The highest correlation with the average temperature was observed in flavanols, which were, in this case, only represented by propelargonidin dimers.

Secondary metabolite biosynthesis can also be affected by light conditions [48]. This was also confirmed by our results, as total phenolic content (89.2%), as well as the contents of hydroxycinnamic acid derivatives (77.5%) and flavanols (76.1%), was correlated with sunshine duration.

#### Individual and Total Anthocyanin Contents

The content of anthocyanins determines the colour of fruit, and they are the main group of phenolic compounds present in strawberries. The colour of fruit is one of the most important quality parameters in fresh fruit but also in processed strawberry products [49]. The colour can be affected by the content but also by the composition of anthocyanins [47]. Additionally, the initial content of anthocyanins is an important quality attribute for the further processing of fruit, as higher contents are desired in order to achieve a product with good colour stability [46,50].

Six different anthocyanins were identified in the strawberry samples (Appendix A). Anthocyanin content differed among the harvests and reached the maximum values in August and September (Table 6). Among the identified anthocyanins, pelargonidin-3-*O*-glucoside showed the highest content. Both light and temperature are important factors that affect the flavonoid pathways in fruits [25,26]. This was partly confirmed by our results, as the average values of total anthocyanin content were negatively correlated with RH (−74.7%) and were positively correlated with sunshine duration (87.1%). However, no significant effects of temperature were observed, which is in contrast to previous studies on strawberries [12,27,28] where higher anthocyanin content was measured in strawberries grown under higher temperatures. Pelargonidin-3-*O*-glucoside, as the major anthocyanin, seemed to be less affected by environmental factors than other anthocyanins did, which is in accordance with a previous study [27] where six short-day strawberry genotypes were studied at different locations. This could explain why the correlation with temperature was not significant. The synthesis of pelargonidin-3-o-glucoside was not affected by the average temperature recorded during the season, while longer periods of sunshine contributed to a higher concentration of this anthocyanin in the fruit. On the other hand, higher RH during the season contributed to lower levels of pelargonidin-3-o-glucoside.

Among the other individual anthocyanins, the highest correlation percentage with the average RH was observed in pelargonidin-3-*O*-rutinoside (−96.4%). Moreover, this anthocyanin also showed a positive correlation with temperature (86.1%). Additionally, pelargonidin-3-(6″ malonyl) glucoside content was also correlated with temperature (72.1%); its content was the highest at the beginning of collection, i.e., at the end of June, and then decreased throughout the season, until it reached its minimum at the end of October. However, not all individual anthocyanins followed the same trend. Pelargonidin-3-*O*-acetylglucoside content showed lower accumulation at higher temperatures and longer sunshine durations. It reached the lowest value at the beginning of the season (at the end of June) and kept increasing, with some variations, until it reached the maximum value at the end of October. These results show that some anthocyanins, such as pelargonidin-3-*O*-rutinoside or pelargonidin-3-*O*-acetylglucoside, can be affected by environmental factors more than others. The synthesis of cyanidin-3-*O*-glucoside and 5-pyranopelargonidin-3-glucoside was not affected by environmental factors. These results suggest that the influence of environmental conditions depends on the composition of anthocyanins in strawberry.

In contrast to these results, the colour parameters (L* and C*) showed a negative correlation with sunshine duration, and the hue angle did not show any correlation with sunshine duration. Previous work [51] found correlations between the colour parameters and the content of anthocyanins, which suggests that there are other factors that influence the colouration of fruit.

### 3.8. Enzyme Activity

Polyphenol oxidase and peroxidase are enzymes responsible for enzymatic browning in fruits, including strawberries [52]. This process can affect the marketability of strawberries. The detected enzyme activity is presented in Figure 4. Peroxidase activity reached its maximum on 15 September, and the lowest values were detected on 2 September and 27 October, when the average temperature also dropped. Regarding polyphenol oxidase activity, the maximum value was observed on 15 September and the lowest was observed on 2 September and 27 October, when the average temperatures dropped. Although there were significant differences among the different collection dates, no significant effects of environmental conditions on enzyme activity were found. Additionally, no correlations between enzyme activity and the contents of anthocyanins and other phenolic compounds were observed (data not shown).

## 4. Conclusions

Strawberry fruit quality is a complex concept that depends on both the physical and chemical attributes of this fruit and that can be affected by environmental factors and the cultivar. Since the importance of everbearing strawberries is growing, it is necessary to study the impact of environmental factors on the fruit quality of these strawberries.

Our study shows that everbearing strawberries can provide fruit fulfilling the quality parameters throughout the season, but it also shows that the composition of strawberries, and the contents of nutrients and bioactive compounds, can vary during the harvest season. While the strawberries largest in size and with a high sugar/acid ratio were harvested under lower temperatures towards the end of the harvest season, in October, strawberries with the highest content of anthocyanins, phenolic compounds and ascorbic acid were collected following longer sunshine durations in August. These changes influenced the nutritional value and taste of the fruit. All the collected strawberries fulfilled the recommended limits regarding diameter and TSSs. However, while under lower temperatures, strawberries have a better taste due to the high sugar/acid ratio, under longer sunshine durations, strawberries contain more health-promoting compounds. To balance these two factors, the optimal harvest time would be when the sunshine duration is still long, to enhance the accumulation of anthocyanins and other phenolic compounds, and when the temperatures start to decrease, to obtain a high sugar/acid ratio, such as the conditions in September in our study. The results of our preliminary research can be the basis for further detailed research on different varieties of everbearing strawberries under changing climatic conditions as well as in several production areas.

## Figures and Tables

**Figure 1 foods-12-01349-f001:**
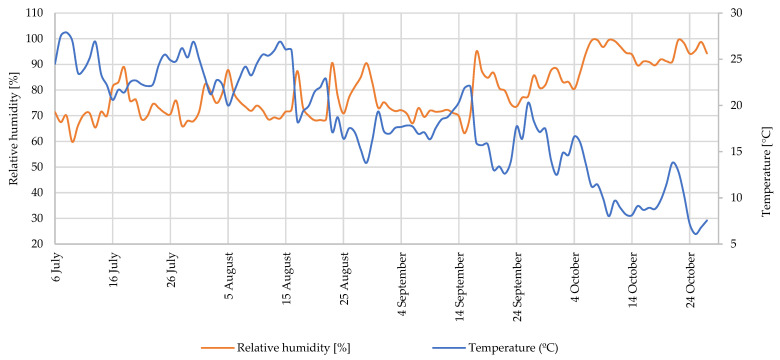
Average daily temperature and relative humidity (RH) values.

**Figure 2 foods-12-01349-f002:**
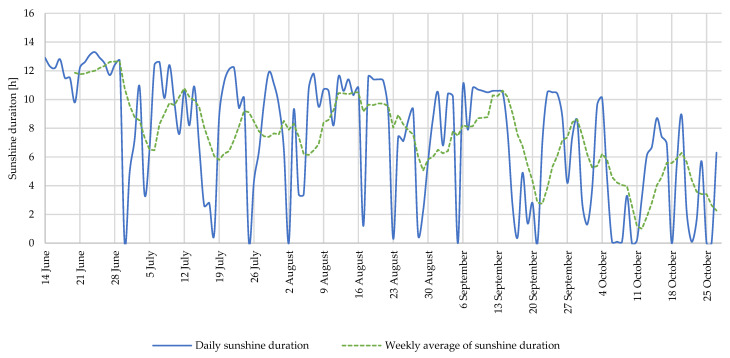
Daily sunshine duration at meteorological station Novo Mesto (ARSO).

**Figure 3 foods-12-01349-f003:**
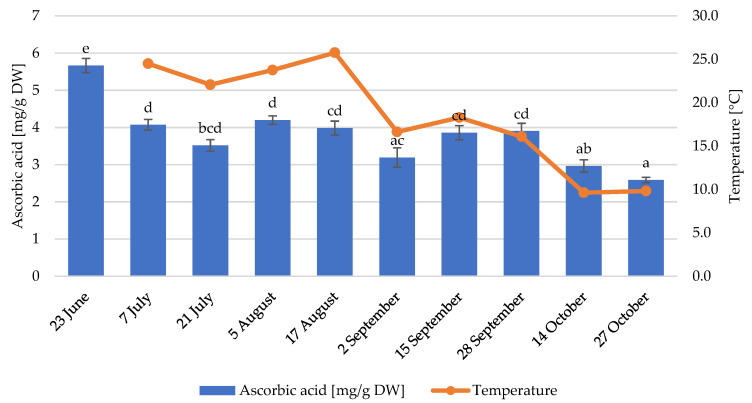
Ascorbic acid contents (mg/g dry weight) and average temperatures of the 7 days before the harvest. Different lowercase letters indicate statistically significant difference between the collection dates (Tukey’s test, *p* < 0.05).

**Figure 4 foods-12-01349-f004:**
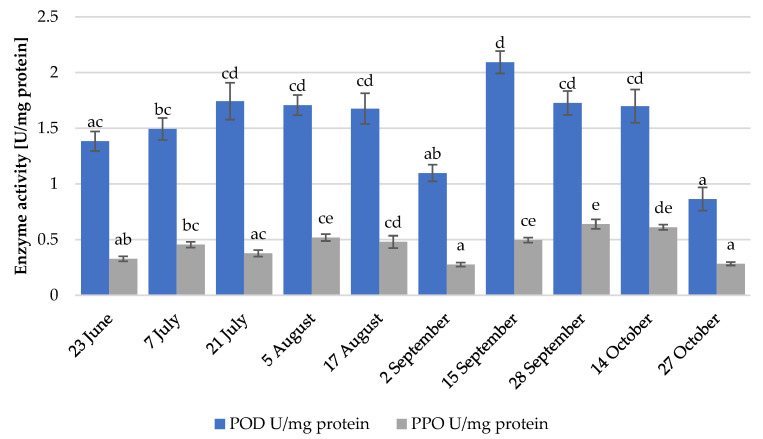
Enzyme activity on different collection dates (POD: peroxidase; PPO: polyphenol oxidase). Different lowercase letters indicate statistically significant difference between the collection dates (Tukey’s test, *p* < 0.05).

**Table 1 foods-12-01349-t001:** Physical parameters measured on fresh fruits at different collection timepoints and Pearson correlation with environmental conditions.

Collection Date	Diameter (mm)	Length (mm)	Weight (g)	Firmness (N)	TSS (°Bx)	Ripening Index	Colour Parameters		
							L*	C*	h°
23 June	35.0 ± 1.0 c	37.1 ± 1.4 de	21.3 ± 2.7 d	0.7 ± 0.2 d	9.1 ± 0.7 ab	0.27 ± 0.02 b	29.8 ± 1.3 ab	36.4 ± 1.8 ac	30.0 ± 1.6 bc
7 July	29.5 ± 0.5 b	32.9 ± 0.9 bcd	12.8 ± 0.7 bc	0.7 ± 0.2 d	10.7 ± 0.6 b	0.26 ± 0.04 ab	28.0 ± 0.4 a	35.6 ± 1.3 ac	26.2 ± 0.4 a
21 July	27.2 ± 0.6 ab	30.6 ± 1.0 ac	10.6 ± 0.9 ab	0.7 ± 0.2 cd	9.3 ± 0.8 ab	0.21 ± 0.04 ab	30.8 ± 0.7 bc	41.7 ± 1.4 d	30.8 ± 0.7 bc
5 August	24.8 ± 0.5 a	28.2 ± 1.7 a	8.7 ± 0.7 a	1.5 ± 0.3 ab	9.2 ± 0.5 ab	0.26 ± 0.03 ab	28.0 ± 0.8 a	34.0 ± 1.1 ab	28.4 ± 2.1 ab
17 August	28.8 ± 0.4 b	29.0 ± 1.1 ab	11.8 ± 0.7 ab	0.8 ± 0.3 cd	9.5 ± 0.8 ab	0.23 ± 0.05 ab	29.3 ± 1.1 ab	33.4 ± 1.2 a	30.5 ± 1.2 bc
2 September	27.9 ± 0.4 b	33.3 ± 1.2 bce	11.7 ± 0.7 ab	1.1 ± 0.2 bcd	9.9 ± 0.5 ab	0.24 ± 0.04 ab	30.4 ± 0.6 bc	37.4 ± 1.0 bc	31.8 ± 1.0 cd
15 September	28.2 ± 0.5 b	32.8 ± 0.9 acd	13.3 ± 1.0 bc	1.3 ± 0.2 ac	9.6 ± 0.4 ab	0.25 ± 0.04 ab	29.3 ± 0.5 ab	37.0 ± 1.2 bc	31.4 ± 1.1 bd
28 September	27.7 ± 0.5 b	34.9 ± 1.4 ce	13.1 ± 0.8 bc	1.0 ± 0.2 bcd	9.3 ± 0.8 ab	0.20 ± 0.03 ab	29.1 ± 0.7 ab	37.9 ± 1.0 c	30.0 ± 1.0 bc
14 October	29.5 ± 0.8 b	37.6 ± 3.9 e	16.1 ± 1.8 c	1.0 ± 0.2 bcd	9.0 ± 0.5 a	0.22 ± 0.03 ab	32.4 ± 0.8 c	42.1 ± 1.9 d	34.5 ± 0.6 d
28 October	33.9 ± 1.0 c	45.2 ± 1.8 f	20.1 ± 1.2 d	1.7 ± 0.2 a	9.9 ± 0.5 ab	0.18 ± 0.02 a	30.5 ± 0.9 bc	42.6 ± 1.2 d	30.8 ± 1.2 bc
Correlation									
RH	ns	80.4% **	74.9% *	ns	ns	ns	78.7% *	81.4% *	ns
Temperature	ns	−86.2% **	−79.9% **	ns	ns	ns	−70.8% *	−77.6% *	−68.3% *
Sunshine duration	ns	ns	ns	ns	ns	ns	−65.8% *	−78.7% **	ns

Different lowercase letters indicate statistically significant difference between the collection dates. Significance of correlation of the environmental conditions is marked as follows: ns: not significant; * and **: significant differences at *p* < 0.05 and *p* < 0.01, respectively.

**Table 2 foods-12-01349-t002:** Pearson correlations between organic acids and sugars, and environmental conditions.

	Ascorbic Acid	Sugars				Organic Acids					Sugar/Acid Ratio
		Sucrose	Glucose	Fructose	Total	Citric	Malic	Shikimic	Fumaric	Total	
RH	−89.6% **	77.5% *	−84.4% **	−85.9% **	ns	−77.4% *	−74.7% *	−74.8% *	−83.6% **	−77.2% *	71.1% *
Temperature	84.8% **	−90.0% **	79.4% *	82.8% **	ns	86.6% **	89.9% **	89.9% **	93.6% **	86.9% **	−77.4% *
Sunshine duration	83.2% **	−67.2% *	76.0% *	78.2% **	ns	ns	ns	ns	ns	ns	ns

ns, not significant; * and **, significant correlations at *p* < 0.05 and *p* < 0.01, respectively.

**Table 3 foods-12-01349-t003:** Organic acid contents on different collection dates.

Collection Date	Organic Acids (mg/g DW)				
	Citric	Malic	Shikimic	Fumaric	Total *
23 June	97.8 ± 6.8 bc	58.7 ± 3.4 cde	0.31 ± 0.03 c	0.27 ± 0.02 d	162.8 ± 9.9 c
7 July	107.2 ± 5.8 cd	61.0 ± 3.0 cde	0.30 ± 0.02 c	0.27 ± 0.01 d	172.9 ± 8.6 cd
21 July	108.4 ± 4.3 cd	59.7 ± 1.9 cde	0.32 ± 0.03 c	0.19 ± 0.01 c	172.2 ± 5.9 cd
5 August	137.6 ± 3.0 e	75.7 ± 2.0 f	0.31 ± 0.02 c	0.19 ± 0.01 c	218.0 ± 3.3 e
17 August	130.3 ± 9.0 de	69.4 ± 5.0 ef	0.31 ± 0.03 c	0.25 ± 0.02 d	204.3 ± 13.9 de
2 September	108.6 ± 5.0 cd	64.4 ± 2.9 df	0.25 ± 0.02 bc	0.17 ± 0.01 c	176.7 ± 6.6 cd
15 September	95.4 ± 1.9 bc	52.6 ± 1.7 bd	0.19 ± 0.01 ab	0.16 ± 0.01 bc	152.2 ± 1.7 bc
28 September	100.3 ± 3.4 bc	51.4 ± 2.1 bc	0.26 ± 0.01 bc	0.15 ± 0.01 ac	156.0 ± 4.9 bc
14 October	79.5 ± 2.7 bc	44.5 ± 2.1 ab	0.17 ± 0.00 ab	0.10 ± 0.00 ab	127.3 ± 3.4 b
28 October	50.6 ± 3.9 a	31.7 ± 1.0 a	0.12 ± 0.01 a	0.08 ± 0.01 a	85.1 ± 4.6 a

* Total organic acid content includes ascorbic acid. Different lowercase letters indicate statistically significant difference between the collection dates (Tukey’s test, *p* < 0.05).

**Table 4 foods-12-01349-t004:** Individual sugar contents, total sugar content and sugar/organic acid ratios on different collection dates.

Collection Date	Sugars (mg/g DW)				Sugar/Acid Ratio
	Sucrose	Glucose	Fructose	Total	
23 June	173.4 ± 2.7 a	399.6 ± 9.5 b	433.3 ± 10.3 d	1006.3 ± 21.2 a	6.3 ± 0.4 bc
7 July	207.0 ± 12.2 ab	370.4 ± 19.5 b	394.5 ± 20.4 cd	971.9 ± 42.4 a	5.7 ± 0.4 ac
21 July	178.7 ± 15.1 a	361.8 ± 23.7 b	382.2 ± 23.5 cd	922.7 ± 60.5 a	5.5 ± 0.5 ac
5 August	175.1 ± 6.9 a	331.6 ± 19.1 ab	352.0 ± 19.4 ad	858.8 ± 44.6 a	3.9 ± 0.2 a
17 August	158.2 ± 10.5 a	358.5 ± 32.3 b	389.9 ± 35.2 cd	906.5 ± 75.3 a	4.4 ± 0.2 ab
2 September	190.6 ± 15.3 ab	352.2 ± 9.8 ab	370.8 ± 9.7 bcd	913.6 ± 34.0 a	5.2 ± 0.2 ac
15 September	225.9 ± 5.8 ac	328.6 ± 7.6 ab	345.3 ± 7.4 ac	899.8 ± 12.7 a	5.9 ± 0.0 ac
28 September	258.8 ± 17.1 bc	363.9 ± 11.4 b	383.4 ± 12.0 cd	1006.0 ± 30.6 a	6.5 ± 0.3 bc
14 October	283.6 ± 17.0 c	273.9 ± 7.3 a	284.0 ± 7.4 a	841.5 ± 23.8 a	6.6 ± 0.2 c
28 October	292.0 ± 30.2 c	270.8 ± 16.9 a	287.1 ± 16.1 ab	849.9 ± 60.9 a	10.2 ± 1.1 d

Different lowercase letters indicate statistically significant difference between the collection dates (Tukey’s test, *p* < 0.05).

**Table 5 foods-12-01349-t005:** Phenolic compound contents on different collection dates and Pearson correlations with average temperature, relative humidity (RH) and sunshine duration.

Collection Date	Phenolics [mg/100 g DW]					
	Total Hydroxycinnamic Acid Der.	Total Flavanols	Total Hydroxybenzoic Acid Der.	Total Flavonols	Total Non-AnthocyaninPhenolic Compounds	Total Phenolic Compounds
23 June	76.1 ± 2.7 ab	32.8 ± 1.8 ce	47.8 ± 2.9 ab	29.0 ± 5.4 a	185.7 ± 11.5 a	709.2 ± 38.1 ac
7 July	94.7 ± 13.0 b	30.9 ± 3.7 cd	58.9 ± 7.4 ab	37.5 ± 7.3 a	222.1 ± 27.3 ac	660.0 ± 52.0 ac
21 July	69.8 ± 7.0 ab	27.3 ± 2.7 ac	67.3 ± 7.0 ab	31.6 ± 6.4 a	196.0 ± 19.8 ab	612.3 ± 67.5 ab
5 August	94.1 ± 6.0 b	43.3 ± 2.1 de	133.8 ± 17.4 c	59.3 ± 5.0 ab	330.5 ± 23.3 c	875.2 ± 24.7 c
17 August	107.0 ± 14.8 b	46.3 ± 5.5 e	130.9 ± 13.0 c	48.9 ± 9.9 ab	333.1 ± 35.8 c	869.2 ± 48.9 c
2 September	78.0 ± 4.7 ab	30.4 ± 2.0 bcd	78.8 ± 11.2 ab	40.9 ± 7.4 a	228.2 ± 19.4 ac	687.3 ± 37.1 ac
15 September	98.1 ± 7.8 b	29.9 ± 2.8 bcd	64.8 ± 9.9 ab	53.7 ± 10.1 ab	246.5 ± 27.5 ac	792.5 ± 75.3 bc
28 September	106.8 ± 7.2 b	30.0 ± 3.6 bcd	94.8 ± 17.8 bc	83.4 ± 18.9 b	314.9 ± 44.8 bc	867.6 ± 55.1 c
14 October	47.1 ± 4.0 a	17.0 ± 0.6 ab	38.9 ± 2.2 a	28.7 ± 5.0 a	131.7 ± 3.9 a	523.0 ± 14.1 a
28 October	54.5 ± 3.8 a	15.1 ± 0.9 a	31.9 ± 2.6 a	36.8 ± 2.9 a	138.3 ± 9.4 a	494.6 ± 22.6 a
Pearson Correlation						
RH	−89.3% **	−81.7% **	ns	ns	−76.7% *	−77.2% *
Temperature	71.9% *	86.3% **	70.4% *	ns	69.4% *	ns
Sunshine duration	77.5% **	76.1% *	ns	ns	ns	89.2% **

Different lowercase letters indicate statistically significant difference between the collection dates (Tukey’s test, *p* < 0.05). ns: non-significant correlation; * and **: significant correlations at *p* < 0.05 and *p* < 0.01, respectively. Total phenolic compounds include the total content of anthocyanins.

**Table 6 foods-12-01349-t006:** Individual anthocyanin contents and total anthocyanin content on different collection dates, and Pearson correlations with average temperature, relative humidity (RH) and sunshine duration.

Collection Date	Anthocyanins (mg/100 g DW)						
	Cyanidin-3-*O*-Glucoside	Pelargonidin-3-*O*-Glucoside	Pelargonidin-3-*O*-Rutinoside	Pelargonidin-3-(6″ Malonyl) Glucoside	5-Pyranopelargonidin-3-Glucoside	Pelargonidin-3-*O*-Acetylglucoside	Total Anthocyanin Content
23 June	12.2 ± 1.6 ab	466.5 ± 28.2 bc	39.0 ± 2.5 b	3.6 ± 0.4 b	0.67 ± 0.10 ab	1.6 ± 0.2 a	523.5 ± 30.3 bc
7 July	15.1 ± 2.2 ab	384.3 ± 37.5 ac	33.6 ± 3.4 b	2.3 ± 0.3 ab	0.90 ± 0.10 b	1.8 ± 0.2 a	437.9 ± 43.3 ac
21 July	11.1 ± 2.4 a	366.4 ± 43.2 ac	33.7 ± 2.7 b	1.9 ± 0.4 ab	0.95 ± 0.24 b	2.2 ± 0.3 a	416.2 ± 49.1 ac
5 August	13.6 ± 1.1 ab	492.9 ± 12.4 bc	34.5 ± 2.4 b	1.3 ± 0.2 a	0.49 ± 0.04 ab	2.0 ± 0.2 a	544.7 ± 13.7 bc
17 August	11.7 ± 1.9 a	485.4 ± 22.9 bc	34.4 ± 3.0 b	2.1 ± 0.1 ab	0.50 ± 0.08 ab	2.0 ± 0.3 a	536.1 ± 27.0 bc
2 September	11.5 ± 1.4 a	414.5 ± 18.2 ac	29.1 ± 1.2 ab	1.1 ± 0.2 a	0.47 ± 0.03 ab	2.5 ± 0.1 a	459.1 ± 20.5 ac
15 September	15.1 ± 2.4 ab	488.4 ± 53.2 bc	37.7 ± 3.3 b	1.3 ± 0.2 a	0.54 ± 0.08 ab	2.9 ± 0.4 ab	546.0 ± 59.1 bc
28 September	20.8 ± 2.7 b	495.2 ± 13.1 c	30.0 ± 0.7 ab	1.3 ± 0.1 a	0.65 ± 0.07 ab	4.6 ± 0.6 bc	552.7 ± 14.5 c
14 October	7.2 ± 0.9 a	355.4 ± 15.9 ab	21.1 ± 0.9 a	1.5 ± 0.9 a	0.41 ± 0.02 a	5.8 ± 1.0 cd	391.3 ± 16.2 ab
28 October	11.1 ± 1.1 a	317.5 ± 12.1 a	19.6 ± 1.9 a	0.7 ± 0.2 a	0.51 ± 0.07 ab	6.9 ± 0.3 d	356.4 ± 14.9 a
Pearson Correlation							
RH	ns	−71.3% *	−96.4% **	ns	ns	87.6% **	−74.7% *
Temperature	ns	ns	86.1% **	72.1% *	ns	−92.3% **	ns
Sunshine duration	ns	84.7% **	90.8% **	ns	ns	−71.8% *	87.1% **

Different lowercase letters indicate statistically significant difference between the collection dates (Tukey’s test, *p* < 0.05). ns, non-significant correlation; * and **, significant correlations at *p* < 0.05 and *p* < 0.01, respectively.

## Data Availability

All data are present in the manuscript and Appendix A.

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
