# Peer review of "Variability in ‘Capri’ Everbearing Strawberry Quality during a Harvest Season"

_foods, 2023, doi:10.3390/foods12061349_

Round 1

Reviewer 1 Report

Reviewer’s comments on the foods-2206848 manuscript:

Lines:

11-12: reorganize the sentence so that the word “strawberry” shows up only once

80: what does “SE” mean?

195: 0,2 to 0.2

356: 77.2 or -77.2?

368-376 (However,…September.) rewrite the denoted sentences, it is not clear and seems that the same statement is repeated

384-385; 429: what about relative humidity?

406: check the correlation coefficients

408: 96.4 or -96.4?

409: check the coefficient

437: check the dates

442: add “Data not shown” or similar

489: journal title?

499, 507-508, 539: delete words “vol” and “page”

521: source?

524, 550: pages?

531: vol?

564-565: journal title abbreviation?

566: is something missing?

Author Response

 The text and the conclusion have been updated based on comments. Dry matter was only measured to be used for calculations and the samples were not further used for any analysis, therefore we chose higher temperatures.

Reviewer 2 Report

This study was determined the optimal harvest conditions for the everbearing cultivar Capri that is commercially grown in Slovenia. Authors decided to study the physical and chemical parameters of strawberry fruit with the focus on sugar, organic acid, the content of anthocyanins and other phenolic compounds, as well as the effect of environmental conditions on them within one harvest season. The environmental conditions (temperature, relative humidity and sunshine duration) were closely monitored throughout the season directly in the field so that the effect of these conditions on fruit quality could be evaluated. The results and ambition of the study are quite interesting and extremely important to consumers. However, a few minor revisions are required and these changes are marked in the attached file. I also suggest that the English linguistic errors of the manuscript be reviewed by a native speaker. I think it can be published after these changes.

Author Response

 The text has been updated as per the comments. The language of the manuscript has also been reviewed and some parts were rephrased. Coefficients were checked and corrected. References were updated based on DOI link using Mendeley and it is updated based on the provided comments.

Reviewer 3 Report

Point 1: The aim of this MS was to determine the optimal harvest conditions for the ever bearing cultivar Capri. So we think the physical and chemical qualities of fruits are the most concerned by consumers. Beside the determined indexes (sugar, organic acid, anthocyanins and phenolic compounds), these qualities also include the vitamin content, mineral element, biological activity, etc. Maybe more quality indexes should be considered in the MS.

Point 2: Additionally, the influence of environmental conditions on fruit growth is very complex. These environmental conditions include water and soil conditions, climate conditions, cultivation and management methods, etc. Did the author consider the influence of these conditions during the experimental design? If not, how to eliminate the influence of other factors?

Point 3: The changes of anthocyanins and polyphenols during strawberry growth were discussed in a large amount in the MS. Hence, we think these two indexes are very important for the objective of this study (to determine the optimal harvest conditions). However, the author did not discuss the relationship between these two indicators and the optimal harvest conditions. What is the relationship between these two indicators and fruit quality and consumer acceptance? Is there a comparative discussion with previous research results? These arguments are closely related to the research objectives of this MS.

Point 4: The keyword selection is unreasonable, sugars and phenolics should be deleted.

Point 5: The author declared that this MS aimed to determine the optimal harvest conditions, but what are the optimal harvest conditions? This question is not answered in the summary or conclusion sections.

Author Response

Thank you for all your valuable comments. The manuscript has now been updated accordingly and you can find the response to your individual points below.

Point 1: The aim of this MS was to determine the optimal harvest conditions for the ever-bearing cultivar Capri. So we think the physical and chemical qualities of fruits are the most concerned by consumers. Beside the determined indexes (sugar, organic acid, anthocyanins and phenolic compounds), these qualities also include the vitamin content, mineral element, biological activity, etc. Maybe more quality indexes should be considered in the MS.

Added notes in the introduction regarding the content of vitamins and minerals The aim of our study was to compare common chemical compounds (sugars, organic acids, phenolic compounds), which represent some of the quality parameters. Ascorbic acid was included in the scope of the study as it is the major micronutrient and an important quality attribute of strawberries. Of course, there remains room for broader analyses, which were not the subject of the research this time.

 Point 2: Additionally, the influence of environmental conditions on fruit growth is very complex. These environmental conditions include water and soil conditions, climate conditions, cultivation and management methods, etc. Did the author consider the influence of these conditions during the experimental design? If not, how to eliminate the influence of other factors?

With the design of the experiment, we minimized the influence of other environmental factors that also affect fruit growth. Fertilization, soil properties and cultivation methods were set and were uniform in several block replications (five blocks). During the duration of the experiment, we monitored the rainfall (on average 19 mm per week, maximum in July – 61 mm in one week) and in times of water shortage (middle of August and middle of September), water was supplied through the irrigation system (drip irrigation system) so that the water supply to plants was optimal. In this way, we tried to ensure uniform conditions for conducting the experiment.

Point 3: The changes of anthocyanins and polyphenols during strawberry growth were discussed in a large amount in the MS. Hence, we think these two indexes are very important for the objective of this study (to determine the optimal harvest conditions). However, the author did not discuss the relationship between these two indicators and the optimal harvest conditions. What is the relationship between these two indicators and fruit quality and consumer acceptance? Is there a comparative discussion with previous research results? These arguments are closely related to the research objectives of this MS.

  • Colour parameters are generally taken into account for fresh consumption, which are closely related to the content of anthocyanins and other phenolic substances. When growing fruit for processing, the phenolic compounds content and, more specifically, the anthocyanin content are particularly important parameters as the initial concentration has a significant effect on the colour stability of the product and consequently consumer acceptance. We added a note with references to the respective results section. During the duration of the experiment, we picked fruits that were technologically ripe in each term. In strawberries, an important ripeness parameter is the red coloring of the fruit, so the values of the color parameter measurements were very similar - the h° angle was between 26 and 34 (although there were differences). The aim was to investigate whether there is an influence of the observed parameters on the content of anthocyanins and other phenolic compounds, even though the color of the fruits is very similar in appearance. The desire in the future is to continue research in the direction of determining these substances with non-destructive methods, such as the determination of phenolic ripeness (in viticulture) and the non-destructive determination of soluble dry matter in fruits.

Point 4: The keyword selection is unreasonable, “sugars” and “phenolics” should be deleted. – Done

Point 5: The author declared that this MS aimed to determine the optimal harvest conditions, but what are the optimal harvest conditions? This question is not answered in the summary or conclusion sections.

  • Added a note to the conclusion section regarding the optimal harvest time. The optimal harvest time would be when the sunshine duration is still long to enhance the accumulation of anthocyanins and other phenolic compounds and the temperatures are starting to decrease to obtain a higher sugar/acid ratio, such as the conditions in September in our study.

Reviewer 4 Report

much more refining scope is lying remaining to address that can be started from title reframing to keywords rearrangement to M&M soundness to R&D discussion.

some of the points are highlighted as comments and suggestions under revision category keeping in mind the journals repo. address them keenly and with full devotion to gain future citation benefits.

Author Response

Thank you for reviewing the article and your comments.

The manuscript has been reviewed. The changes include: 

  • Rephrasing of the article title
  • Rephrasing of the abstract
  • Removal of some of the keywords based on reviewers’ suggestions
  • Addition of more details and references on the bioactive compounds present in strawberries in the Introduction
  • Adding details about the ripeness of the fruit and replicates done in the Methods section
  • Adding details and references on the importance of phenolic compounds in the Results section
  • Adding details to the conclusion to clarify the optimal harvest time

We believe these changes will improve the understanding of the readers and add to the better interpretation of the results. 

If you have additional and specific suggestions, please let us know and we will take them into consideration.

Round 2

Reviewer 3 Report

I still think this MS does not meet the standard for publication in Foods. There are many defects in the experimental design of the MS.

Author Response

I believe there was a misunderstanding. The previous revised version was uploaded before your comments were received and therefore were not implemented yet. Now the current version should address your comments and further details were provided in the Round 1 response.